# *Artemisia anomala* Herba Alleviates 2,4-Dinitrochlorobenzene-Induced Atopic Dermatitis-Like Skin Lesions in Mice and the Production of Pro-Inflammatory Mediators in Tumor Necrosis Factor Alpha-/Interferon Gamma-Induced HaCaT Cells

**DOI:** 10.3390/molecules26175427

**Published:** 2021-09-06

**Authors:** Ju-Hye Yang, Kwang-Youn Kim, Young-Woo Kim, Kwang-Il Park

**Affiliations:** 1Korean Medicine (KM) Application Center, Korea Institute of Oriental Medicine, 70 Cheomdan-ro, Dong-gu, Daegu 41062, Korea; jjuhye@kiom.re.kr (J.-H.Y.); lokyve@kiom.re.kr (K.-Y.K.); 2School of Korean Medicine, Dongguk University, Gyeongju 38066, Korea; 3Department of Veterinary Physiology, College of Veterinary Medicine, Gyeongsang National University, Jinju 52828, Korea

**Keywords:** *Artemisia anomala* S. Moore, atopic dermatitis, dinitrochlorobenzene, keratinocytes

## Abstract

*Artemisia anomala* S. Moore is a perennial herbaceous plant classified as Asteraceae of the genus Artemisia. Many species of *Artemisia* have been used as medicinal materials. *Artemisia anomala* S. Moore has been widely used in China to treat inflammatory diseases. However, the mechanism of its action on the keratinocyte inflammatory response is poorly understood. Here, we investigated the anti-inflammatory reaction of *Artemisia anomala* S. Moore ethanol extract (EAA) using human keratinocyte (HaCaT) cells, which involved investigating the nuclear factor kappa B (NF-κB), signal transducer, and activator of transcription-1 (STAT-1), as well as mitogen-activated protein kinase (MAPK) signaling pathways and atopic dermatitis-like skin lesions in mice. We elucidated the anti-inflammatory effects of EAA on tumor necrosis factor-α/interferon-γ (TNF-α/IFN-γ)-treated human keratinocyte cells and 2,4-dinitrochlorobenzene (DNCB)-induced atopic dermatitis (AD)-like mice. The levels of chemokines and cytokines (IL-8, IL-6, TARC, and RANTES) were determined by an enzyme-linked immunosorbent assay. The NF-κB, STAT-1, and MAPK signaling pathways in HaCaT cells were analyzed by western blotting. Thickening of the mice dorsal and ear skin was measured and inflammatory cell infiltration was observed by hematoxylin and eosin staining. Results showed that EAA suppressed IL-8, IL-6, TARC, and RANTES production. EAA inhibited nuclear translocation of NFκB and STAT-1, as well as reduced the levels of phosphorylated ERK MAPKs. EAA improved AD-like skin lesions in DNCB-treated mice. These findings suggest that EAA possesses stronger anti-inflammatory properties and can be useful as a functional food or candidate agent for AD.

## 1. Introduction

*Artemisia anomala* S. Moore is called “Yu-Gi-no” in Korean medicine and ”Nan-Liu-Ji-Nu” in Chinese medicine, and is a perennial herbaceous plant classified as the *Artemisia* genus in the Compositae family [1]. *Artemisia anomala* S. Moore has long been used in the treatment of diseases through modern medicine and traditional medicine, such as for dissipated liver function caused by hepatitis, fever, and inflammation in China, Japan and Korea [2]. In addition, constituents of *Artemisia anomala* S. Moore, such as quercetin, apigenin, and tricin, were reported for anti-inflammation, anti-oxidant and anti-cancer effects. However, the mechanisms of *Artemisia anomala* S. Moore in atopic dermatitis (AD) in terms of its therapeutic effect are not clear [3,4].

Skin, the most external organ of the human body, plays a central role in maintaining homeostasis between the body and the external environment, and protects the body from the penetration of external pathogens such as chemical and physical damage. It is composed of three main layers including the dermis, epidermis, and subcutaneous fat layer [5,6]. The epidermis in particular comprises various cell types, such as keratinocytes, langerhans cells, merkel cells, and melanocytes. Among them, the epidermal keratinocytes, the predominant type, account for approximately 95% of the total cell population [7,8]. They produce various cytokines and chemokines, such as IL-8, IL-6, RANTES, and TARC, upon invasion of external skin pathogens. These factors lead to skin inflammation through an increase in immunocyte infiltration at the site of inflammation in the skin [9,10,11,12]. Previous studies have reported that transcription factors, such as the nuclear factor kappa-light-chain-enhancer of activated B cells (NF-kB), signal transducer, and activator of transcription-1 (STAT-1), as well as mitogen-activated protein kinase (MAPK), play important roles in the production of cytokines and chemokines in skin inflammation [13,14,15]. Therefore, inhibition of pro-inflammatory cytokines and chemokine production through the regulation of transcription factors, including NF-kB, STAT-1, and MAPKs in epidermal keratinocytes, can be an adequate strategy for inflammatory skin diseases.

The present study aimed to investigate the anti-inflammatory effects of the *Artemisia anomala* S. Moore ethanol extract (EAA) by inhibiting tumor necrosis factor-α/interferon-γ (TNF-α/IFN-γ)-induced ERK and NFκB signaling in HaCaT cells, and by improving the skin conditions in 2, 4-dinitrochlorobenzene (DNCB)-induced atopic dermatitis-like lesions in the mouse model. Additionally, the results of this study will contribute in evaluating the potential of EAA as an anti-inflammatory ingredient towards protecting the skin from diseases such as AD.

## 2. Results

### 2.1. In Vitro Cell Cytotoxicity in Keratinocytes

The in vitro cell cytotoxicity of EAA was observed by MTT assay. EAA showed less than 10% cytotoxicity and about 94% and 92% cell viability at 100 and 200 µg/mL, respectively (Figure 1).

### 2.2. EAA Suppresses the Production of AD-Related Cytokines and Chemokines in Keratinocytes

To investigated the anti-atopic dermatitis effects and mechanism of EAA, the experiments were performed by co-stimulation of TNF-α and IFN-γ in human keratinocytes. At various concentrations, excluding cytotoxicity, EAA was pretreated for 1 h and then treated for 24 h with stimulants (TNF-α and IFN-γ). The cytokines and chemokines secreted in the cell culture were then analyzed using ELISA in the supernatants. As shown in Figure 2, EAA significantly reduced the production of cytokines and chemokines, such as RANTES, IL-8, TARC, and IL-6, in HaCaT cells co-stimulated by TNF-α and IFN-γ in a concentration-dependent manner. EAA treatment significantly decreased the productions of RANTES, IL-8, TARC, and IL-6 to 21%, 22.5%, 24.5%, and 8.2% at 50 µg/mL, respectively.

### 2.3. EAA Inhibits NFκB and STAT-1 Activation in Keratinocytes

NF-κB was controlled for by the inhibition of κB (IκB) proteins. IκB-α forms a complex with NF-κB, blocks the activation of NF-kB, and activates NF-κB upon isolation. IκB-α is present in the cytoplasm and eventually degraded. As shown in Figure 3a,c, IκB-α was remarkably degraded by stimulant treatment with TNF-α/IFN-γ and pretreatment with EAA significantly recovered this degradation. Furthermore, the levels of phosphorylated STAT-1 significantly reduced after the treatment with EAA.

### 2.4. EAA Inhibits MAPKs in Keratinocytes

TNF-α/IFN-γ stimulation increased the expressions of MAPKs in HaCaT cells (Figure 3b,d). In the EAA-treated group, MAPKs reduced the TNF-α/IFN-γ-induced increase of ERK in a concentration-dependent manner, whereas the levels of phosphorylated p38 and JNK did not change. Furthermore, the levels of total MAPKs were not affected.

### 2.5. EAA Reduces the Symptoms of AD in Mice

To evaluate the effect of EAA on DNCB-induced AD in mice, we used DNCB to repeatedly expose the dorsal and ear skin areas of BALB/c mice. The results show that the skin condition significantly improved and the epidermal thickness of the dorsal skin reduced in EAA-treated mice compared to the control group. EAA significantly reduced the ear skin thickness that was increased by DNCB in a dose-dependent manner (Figure 4 and Figure 5a). Furthermore, we analyzed the size of the spleen, which are the main organ in the inflammatory reaction. The spleen plays a critical role in hematopoiesis and immune response. Splenomegaly was used as an indicator of inflammatory reaction. As shown in Figure 5b, EAA reduced the size of the spleen in the DNCB-induced AD-like skin lesions of the mice. In addition, EAA significantly reduced the production of cytokines such as TNF-α and IFN-γ that were increased by DNCB in a dose-dependent manner (Figure 6). Based on these results, EAA has favorable effects that can reduce the symptoms of AD-like skin lesions in mice.

### 2.6. HPLC Analysis of EAA

According to the maximum absorption of the standards, the ultraviolet (UV) detector was set at 280 nm for the HPLC analysis of three standard compounds, including quercetin (1), apigenin (2), and tricin (3). The HPLC chromatograms of the standard mixture and EAA extract are presented in Figure 7. The mixed standards were indicated at the retention time of 29.8 (1), 32.4 (2), and 33.1 min (3). Under the same conditions, the components of EAA were observed at 29.9 (1), 32.4 (2), and 33.1 min (3) by comparing UV spectral data with the standard compounds. The amounts of the three components in EAA were analyzed by applying regression equations calculated from the calibration curves. The quantitative analysis of the constituents in EAA are summarized in Table 1.

## 3. Discussion

Traditional Chinese medicines have been used for various diseases because they possess numerous pharmacological activities [16]. Therefore, there are useful dietary sources with health benefits. Among them, *Artemisia anomala* S. Moore has been evaluated for its various pharmacological effects but the precise mechanism of its therapeutic effect on AD remains unclear [3,4]. In this study, we investigated the effects of EAA in the DNCB-induced AD mouse model and TNF-α/IFN-γ-stimulated HaCaT cells.

Keratinocytes are critical elements in the regulation of skin pathology in AD. Keratinocytes are located in the outermost part of the epidermis of the skin and play an important role in the pathogenesis of inflammatory skin disease, in addition to secreting pro-inflammatory mediators as a cellular source of risk signals. Activated keratinocytes can induce the upregulation of pro-inflammatory chemokines and chemokines, including interleukin-6 (IL-6), interleukin-8 (IL-8), regulated upon activation (RANTES; CCL5), and TARC; CCL17) [17]. These skin-related cytokines and chemokines are involved in the exacerbation of AD by selectively controlling the migration of immune cells to the skin lesion site and stimulating an inflammatory response [18]. In this experiment, we elucidated that EAA inhibits the production of pro-inflammatory cytokines and chemokines in TNF-α/IFN-γ-stimulated keratinocytes. In this experiment, we elucidated that EAA inhibits the production of pro-inflammatory cytokines and chemokines, such as IL-6, IL-8, RANTES, and TARC in TNF-α/IFN-γ-stimulated keratinocytes, consistent with previously reports [17,18]. 

It is well known that the activation of the degradation of IκB-α, STAT-1, and MAPK in keratinocytes are associated with AD, and keratinocytes stimulated with TNF-α/IFN-γ are widely used for verification of the in vitro efficacy of anti-atopic dermatitis [19,20]. In the current study, EAA suppressed the phosphorylation of STAT-1, ERK and the degradation of IκB-α in TNF-α/IFN-γ-stimulated keratinocytes.

AD, as an allergic inflammatory skin disease, has high prevalence and recurrence rates resulting from immune system dysregulation [21]. In the acute phase, AD leads to the thickening of the epidermis and dermis; the infiltration of various immune cells, including helper T (Th) cells, mast cells, and eosinophils; and the increase in immune cells-associated cytokines, such as through the overexpression of Th2 cytokines and IgE production [22,23,24]. IgE hyper-production has been associated with the pathogenesis of AD and the concentration of serum IgE is promoted in patients with AD [13,25]. Steroid therapy plays a pivotal role in the treatment of AD but its long-term administration is prohibited due to critical side effects [26,27,28]. In the present study, we found that EAA significantly reduced AD symptoms in mice. We observed that EAA treatment reduced DNCB-induced AD-like skin lesions on the dorsal and ear skin in mice. Furthermore, EAA improved the splenomegaly, which is an indicator of inflammation (Figure 4 and Figure 5). In addition, EAA reduced the production of pro-inflammatory cytokines, including TNF-α and IFN-γ, in DNCB-induced mouse skin (Figure 6).

## 4. Materials and Methods

### 4.1. Reagents and Cell Culture 

*Artemisia Anomala* Herba was obtained from the Yeongcheon Oriental Herbal Market (Yeongcheon, Korea). EAA was extracted with ethanol as in a previously reported recipe [1]. ELISA kits for IL-8 (#431501), RANTES (#440804), thymus and activation regulatory chemokines (TARC, #441104), IL-6 (#430501), recombinant human tumor necrosis factor alpha (TNF-α), and the interferon gamma (IFN-γ) were purchased from BioLegend (San Diego, CA, USA). Antibodies used in the western blot analyses were purchased from Cell Signaling Technology (Boston, MA, USA).

### 4.2. Cell Culture

Human keratinocytes HaCaT cells were maintained in DMEM supplemented with 10% FBS (Lonza, Basel, Switzerland), 100 IU/mL penicillin, and 100 μg/ mL streptomycin at 37 °C in 5% CO_2_. Confluent cells were split (1:5–1:8 ratio) by trypsinization and used at the third/fourth passage after thawing. 

### 4.3. Cell Viability Test

Cell viability was tested using the MTT assay as described previously [1]. Briefly, cells were allowed to attach to plates that were treated with 1, 10, 50, 100, and 200 μg/mL EAA for 24 h. MTT solutions were then added to each well and the cells were incubated for an additional 2 h. The resulting formazan was dissolved using DMSO and the optical density at 570 nm on the VERSAmax microplate reader (Molecular Devices, Sunnyvale, CA, USA).

### 4.4. Enzyme-Linked Immunosorbent Assay (ELISA)

HaCaT cells were seeded in 6-well plates at a density of 5 × 10^5^ cells/well and cultured overnight at 37 °C in a 5% CO2 incubator. The cells were stimulated to 10ng/mL of TNF-α and 10 ug/mL of IFN-γ for 1 h and cultured for another 24 hr. The supernatant was collected and cell debris were removed by centrifugation at 1000× *g* for 10 min. Chemokines and cytokines were analyzed using the following ELISA kits according to the manufacturer’s instructions.

### 4.5. Western Blot Analysis

HaCaT cells were seeded in 6-well plates at a density of 5 × 10^5^ cells/well and pretreated with the indicated concentrations of EAA for 2 h, followed by stimulation with TNF-α/IFN-γ (each 10 ng/mL) for 30 min. The total protein was extracted by RIPA lysis buffer (Millipore, Billerica, MA, USA) containing the protease and phosphatase inhibitor cocktail (Roche, Basel, Switzerland). The PierceTM BCA protein assay kit (Thermo Fisher scientific, Waltham, MA, USA) was applied to determine the concentration of total proteins. Protein samples (30 μg) were separated by sodium dodecyl sulfate-polyacrylamide-gel electrophoresis and then electrotransfered to Polyvinylidene Difluoride membranes. The membranes were blocked with 3% bovine serum albumin and incubated with primary antibodies at 4 °C overnight. The blots were subsequently incubated with HRP-conjugated secondary antibodies at room temperature for 1 h. Specific proteins were detected using the ClarityTM western ECL substrate (Bio-Rad, Hercules, CA, USA). The signals were finally captured and the intensity of proteins on the bands was quantified using the Image J software (National Institutes of Health, Bethesda, MD, USA).

### 4.6. Atopic-like Dermatitis Mouse Model

Five-week-old male BALB/c mice were purchased from Samtako BioKorea (Osan, Korea). The atopic-like dermatitis model was performed as in a previously reported method [20]. All procedures for the animal study were approved by the Korea Institute of Oriental Medicine Institutional Animal Care and Use Committee (KIOM-IACUC; D-17-013) and were conducted in accordance with US guidelines (NIH publication #83–23, revised in 1985). Sensitization protocols were carried out as described previously [29].

### 4.7. Histopathological Observation

In order to reduce the error, the designated researcher continuously measured. Ear thickness and spleen size were evaluated by a micrometer (Mitutoyo, Kawasaki, Japan). For the histological analysis, dorsal skins were fixed in 4% paraformaldehyde and embedded in paraffin wax. The skin tissue sections were stained with hematoxylin and eosin (H&E) staining to measure changes in the epidermal thicknesses. All skin images were acquired under Nikon Eclipse Ti microscope (Nikon, Tokyo, Japan). Representative sections of the epidermis and dermis were collected by H&E staining (magnification ×400, scale bar: 100 μm). The thickness of the skin epidermis was measured from each of the five locations using Image J software (National Institute of Health, Starkville, MD, USA).

### 4.8. Isolation of RNA, cDNA Synthesis, and Real-Time Reverse Transcription-Polymerase Chain Reaction (RT-PCR) Analysis

Total RNA was extracted using the Trizol reagent (Invitrogen Life Technologies, CA, USA) following the manufacturer’s instructions. The total RNA was quantified by measurement using a nanodrop2000 spectrophotometer (Thermo Fisher scientific, Waltham, MA, USA). The total RNA was transformed into cDNA using the ReversTra Ace^TM^ qPCR RT master mix (TOYOBO, Osaka, Japan) according to the following protocol: incubate at 37 °C for 20 min; incubate at 50 °C for 5 min; heat to 98 °C for 5 min; and store the reacted solution at 4 °C. Real-time RT-PCR was performed on a CFX384 Real-time system (Bio-rad, CA, USA) by using the THUNDERBIRD^TM^ SYBR^TM^ qPCR Mix (TOYOBO, Osaka, Japan). Primers for PCR amplification were as follows: GAPDH: 5′-AAC GAC CCC TTC ATT GAC-3′/5′-TCC ACG ACA TAC TCA GCA C-3′; IFN-γ: 5′-AGA GGA TGG TTT GCA TCT GGG TCA-3′/5′-ACA ACG CTA TGC AGC TTG TTC GTG-3′; and TNF-α: 5′- ATG AGC ACA GAA AGC ATG AT-3′/5′-TAC AGG CTT GTC ACT CGA AT-3′. The RT-PCR reactions were cycled 40 times with denaturation (95 °C, 15 s) and annealing (60 °C, 1 min). The fold change in the target gene expression relative to the control was normalized to β-actin using the 2^−ΔΔCt^ method. Real-time RT-PCR was contributed to detect the effects of relevant inflammatory factors on mRNA.

### 4.9. Conditions of Chromatographic

The HPLC data were acquired by the SPD 20A system (Shimadzu Co., Nakagyo-ku, Kyoto, Japan). The solvents used were 0.1% HCOOH and MeCN. Analysis was performed using a gradient system with a flow rate of 1.0 mL/min and a YMC gel ODS A302 column (length, 250 mm; inner diameter, 4.6 mm; particle size, 5 µm; YMC CO., LTD., Shimogyo, Kyoto, Japan) was used. The solvent conditions used in the mobile phases were 0–20 min at 8–20%, 20–30 min at 40%, 30–40 min at 70%, and 40–55 min at 100%. Samples were analyzed at a wavelength of 360 nm at 25 °C. The content of the peak area was obtained from the UV and standard material. Polyphenol samples were quantified in a LC-UV chromatogram with three selected standards. Quantification of the polyphenolic compound detected in a sample of *Artemisia anomala* was conducted using HPLC 360 nm. The quantification performance was validated in terms of linearity and content. A calibration curve was established for each of the standards using five concentration levels (*n* = 5; 62.5, 125, 250, 500, and 1000 µg/mL) and the polyphenol content was measured in terms of peak area ratios with the analyte vs. analyte concentrations using 1/x(x, concentration) weighted linear regression (*n* = 5). The plant’s polyphenolic compounds were routinely quantified using standard curves of structurally related compounds but were quantified using standards that match each compound.

### 4.10. Statistical Analysis

Each experiment was performed at least in triplicates. Data were analyzed by the software GraphPad (Prism 5.0, San Diego, CA, USA). Results were expressed as mean ± S.E and evaluated using Student’s *t*-test or analysis of variance. The statistical difference was defined at a *p*-value of <0.05 (* *p* < 0.05, ** *p* < 0.01, and *** *p* < 0.001). All experiments were performed at least in triplicates.

## 5. Conclusions

This study shows that EAA inhibited the production of cytokines and chemokines, which are involved in the development and relapse of atopy dermatitis. EAA showed anti-inflammatory effects through a mechanism of down-regulated phosphorylation of ERK MAPK and through inhibiting nuclear translocation of NFκB and STAT-1. Moreover, EAA improved the condition of AD-like skin lesions in the mice model treated with DNCB. These findings suggest that EAA has stronger anti-inflammatory properties. Therefore, EAA can be useful as a functional food or candidate agent for AD.

## Figures and Tables

**Figure 1 molecules-26-05427-f001:**
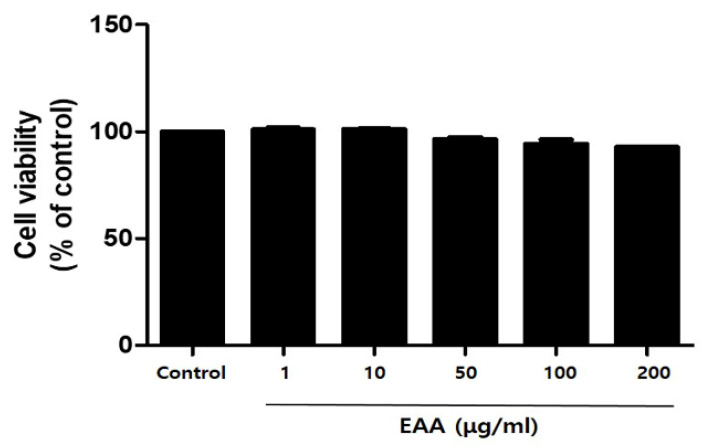
Cytotoxicity of EAA in HaCaT cells. Cytotoxic effects of EAA at different concentrations on HaCaT cells were determined using MTT assays. Results are expressed as a mean percentage in the treated cells compared to the control (solvent) ± S.E. of three independent experiments performed in triplicates.

**Figure 2 molecules-26-05427-f002:**
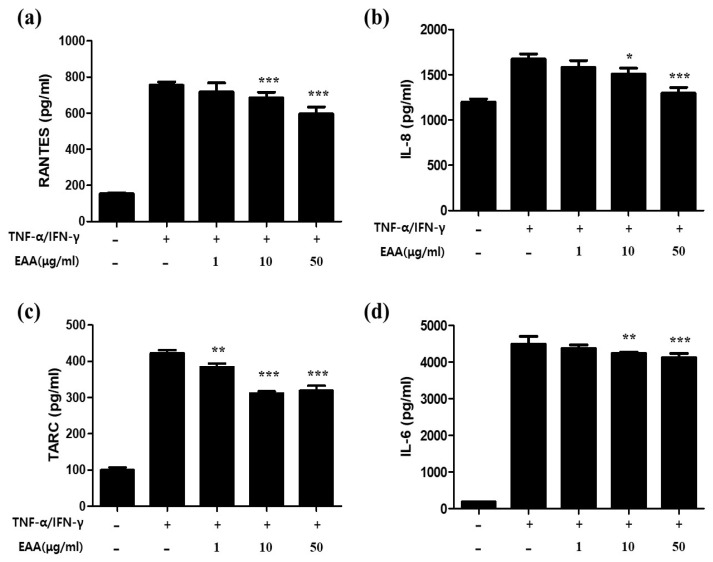
Inhibitory effects of EAA on TNF-α/IFN-γ -induced pro-inflammatory cytokine and chemokine production in HaCaT cells. Cells were pretreated with the indicated concentrations of EAA (1, 10, and 50 μg/mL) for 1 h and then were stimulated with TNF-α/IFN-γ (10 ng/mL each) for 24 h. Production of (**a**) RANTES, (**b**) TARC, (**c**) IL-6, and (**d**) IL-8 was determined using culture supernatants of HaCaT cells. Results are expressed as a mean percentage in the treated cells compared to the control (solvent) ± S.E. of three independent experiments performed in triplicates. * *p* < 0.05, ** *p* < 0.01, and *** *p* < 0.001 compared with the TNF-α/IFN-γ-treated control group.

**Figure 3 molecules-26-05427-f003:**
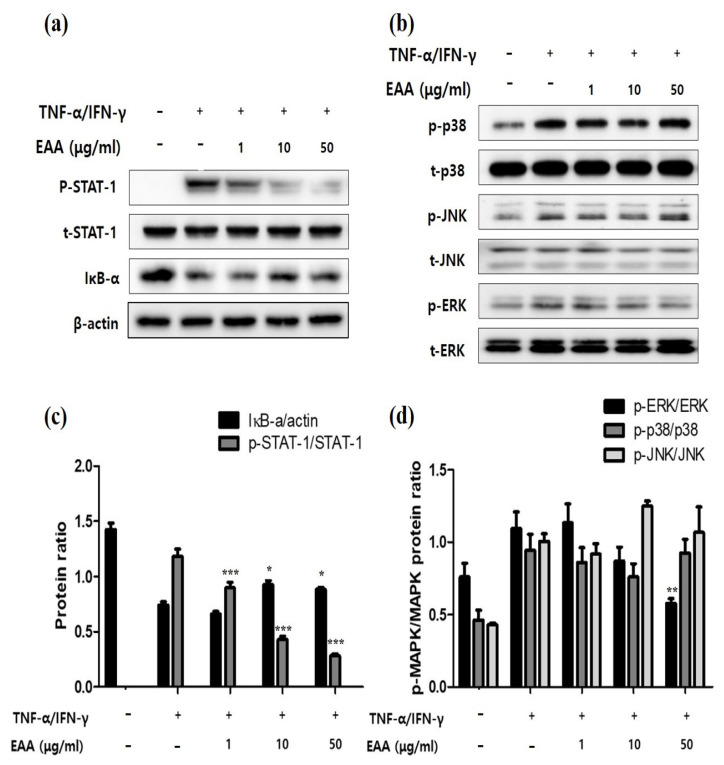
Inhibitory effects of EAA on MAPKs and STAT-1 signaling pathways in TNF-α/IFN-γ-stimulated HaCaT cells. HaCaT cells were pre-incubated with EAA for 1 h and then exposed to TNF-α/IFN-γ (10 ng/mL each) for 30 min. (**a**,**c**) Effects of EAA on STAT-1 activation and IκB-α degradation. (**b**,**d**) Effects of EAA on MAPK phosphorylation in HaCaT cells. Levels of phosphorylation and expression were determined by western blotting using the indicated antibodies. Blots were normalized to the total protein or β-actin. Results are expressed as a mean percentage in the treated cells compared to the control (solvent) ± S.E. of three independent experiments performed in triplicates. * *p* < 0.05, ** *p* < 0.01, and *** *p* < 0.001 compared with the TNF-α/IFN-γ-treated control group.

**Figure 4 molecules-26-05427-f004:**
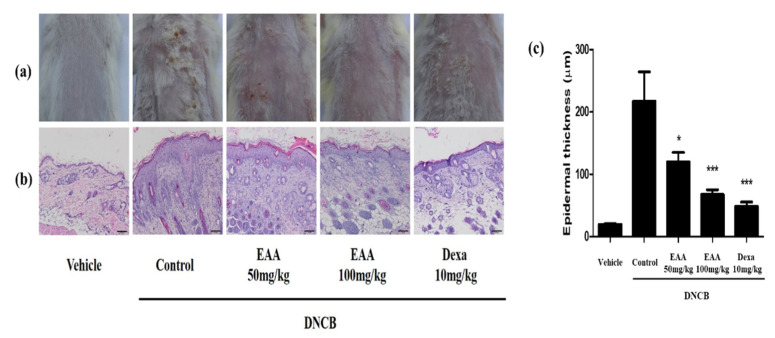
Inhibitory effects of EAA on DNCB-induced AD in the in vivo model. Representative photographs of dorsal skin lesions and histopathological observations in DNCN-induced AD mice with the vehicle (solvent), control (DNCB-induced AD), EAA (50 and 100 mg/kg), and dexamethasone (10 mg/kg) via oral administration. (**a**) Dorsal skin photographs taken at the end of the experiment showing AD-like skin lesions. (**b**) The AD dorsal skin lesions were sectioned and stained with H&E (magnification ×40, scale bar: 100 μm). (**c**) Measurement of the epidermal thickness. Bars with different letters indicate statistically significant differences at * *p* < 0.05 and *** *p* < 0.001 compared with the control group (solvent).

**Figure 5 molecules-26-05427-f005:**
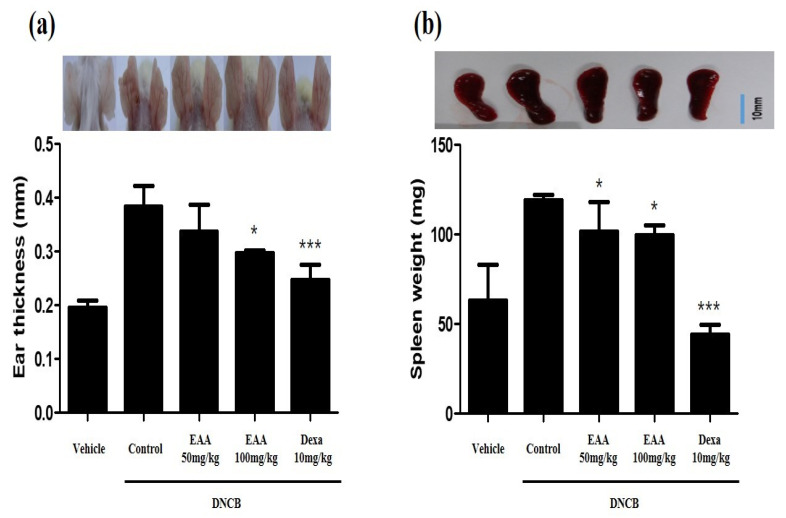
Effects of EAA on ear thickness and spleen hypertrophy. (**a**) Photographs of ear skin lesions and measurements of ear thickness in DNCB-induced AD mice. Ear skin thickness measured with the help of a digital caliper. (**b**) Photographs of the spleen and spleen weight in DNCB-induced AD mice. Where applicable, data are presented as the mean ± S.E. (*n* = 4). Bars with different letters indicate statistically significant differences at * *p* < 0.05 and *** *p* < 0.001 compared with the control group (solvent).

**Figure 6 molecules-26-05427-f006:**
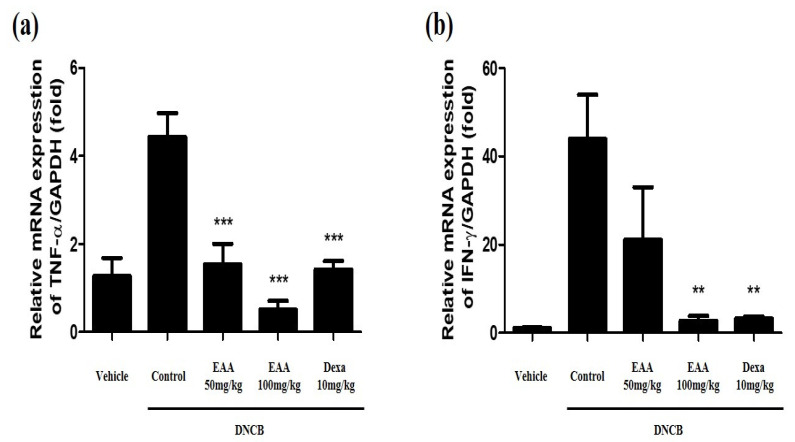
Inhibitory effects of EAA on the DNCB-induced pro-inflammatory cytokine production on skin tissue in the in vivo model. Production of (**a**) TNF-α and (**b**) IFN-γ was determined using real-time RT-PCR. Where applicable, data are presented as the mean ± S.E. (n = 4). Bars with different letters indicate statistically significant differences at ** *p* < 0.01 and *** *p* < 0.001 compared with the control group (solvent).

**Figure 7 molecules-26-05427-f007:**
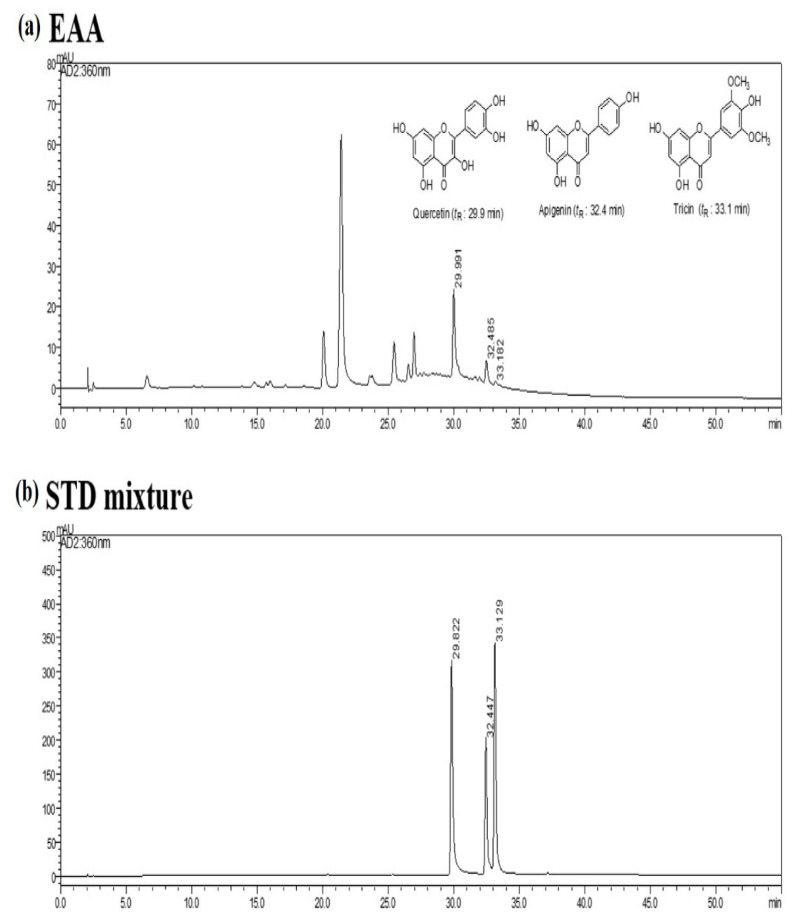
Identification and measurement of the components of EAA by HPLC analysis. HPLC chromatograms of EAA (**a**) and the three standard compound mixtures (**b**) at 280 nm, including quercetin, apigenin, and tricin.

**Table 1 molecules-26-05427-t001:** Quantitative analysis of the constituents in EAA.

		Regression Equation	Linear Range	Content
Compounds	*t*_R_ (min)	(y = ax + b, *R*^2^)	(mg/mL)	(mg/g)
Quercetin	29.9	y = 0.494179x + 4.1, 0.9995	10–200	14.1 ± 0.7
Apigenin	32.4	y = 0.124202x + 2.8, 0.9996	10–200	5.2 ± 0.4
Tricin	33.1	y = 0.932455x + 3.8, 0.9998	10–200	0.4 ± 0.05

## Data Availability

The data presented in this study are available upon request from the corresponding author.

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
