# Peer review of "Artemisia anomala Herba Alleviates 2,4-Dinitrochlorobenzene-Induced Atopic Dermatitis-Like Skin Lesions in Mice and the Production of Pro-Inflammatory Mediators in Tumor Necrosis Factor Alpha-/Interferon Gamma-Induced HaCaT Cells"

_molecules, 2021, doi:10.3390/molecules26175427_

Round 1
Reviewer 1 Report
In the present manuscript, the authors have elucidated how Artemisia anomala S. Moore ethanol extract (EAA) act as an anti-inflammatory agent using human keratinocyte cells. Further, they also confirmed the EAA alleviates the symptoms of DNCB-induced atopic dermatitis-like model mice. Although the manuscript seems to be well written and the works may be performed through appropriate procedures with satisfied results, however, the authors have to provide information regarding following points for satisfying the criteria for the publication.
Major Comments
- For the HPLC analysis, I recommend the authors to perform LC-MS to obtain a rigid evidence, if possible (just suggestion).
- Are there any data about profiles of cytokine expression on animal experiment? Certainly, histopathological observations and the reduction of ear thickness well support the effect of the EAA scientifically. However, it is not necessarily the case that the observed changes of signaling cascade on in vitro cell model correspond with those on the animals. Since the authors have orderly elucidated the mechanism of anti-inflammatory effect on the EAA, might as well confirm a part of signals on animal experiment (e.g. the changes of inflammatory cytokines and related proteins in serum or target organisms), resulting the enough evidence to prove the proposed mechanism. Conversely, without those information, the authors could not associate in vitro and in vivo experiments.
Minor Comments
- Artemisia anomala should be described in italics?
- In Fig. 3, the description about EAA dose (second rows in each) should be changed from “-, +, 1, 10, 50” to “-, -, 1, 10, 50”.
- Since enough margin is provided, the authors should enlarge Fig. 6 for more easy to see.
- If the authors have the approved number of the animal experiment, please provide it in the manuscript.
Author Response
Major Comments
1. For the HPLC analysis, I recommend the authors to perform LC-MS to obtain a rigid evidence, if possible (just suggestion).
: In general, quantitative and qualitative evaluation of compounds in studies using natural products uses HPLC analysis.In this study, the compounds contained in Artemisia anomala S. Moore ethanol extract (EAA) were identified through HPLC analysis.
2. Are there any data about profiles of cytokine expression on animal experiment? Certainly, histopathological observations and the reduction of ear thickness well support the effect of the EAA scientifically. However, it is not necessarily the case that the observed changes of signaling cascade on in vitro cell model correspond with those on the animals. Since the authors have orderly elucidated the mechanism of anti-inflammatory effect on the EAA, might as well confirm a part of signals on animal experiment (e.g. the changes of inflammatory cytokines and related proteins in serum or target organisms), resulting the enough evidence to prove the proposed mechanism. Conversely, without those information, the authors could not associate in vitro and in vivo experiments.
: We investigated the production of pro-inflammatory cytokines TNF-α and IFN-γ using skin tissue from a DNCB-induced atopic dermatitis-like mouse model by using real-time RT-PCR and added them to the results section (Figure 6). In addition, the method of real-time RT-PCR was added to the Materials and Methods section. The corrected part is expressed the red text in manuscript.
Result section
Figure 6. Inhibitory effects of EAA on DNCB-induced pro-inflammatory cytokine production on skin tissue in vivo model. Production of (a) TNF-α, and (b) IFN-γ was determined by using real-time RT-PCR. Where applicable, data are presented as the mean ± S.E. (n = 4). Bars with different letters indicate statistically significant difference at *p < 0.05 and ***p < 0.001 compared with the control group (solvent).
Discussion section
“Also, EAA reduced the production of pro-inflammatory cytokine including TNF-α and IFN‑γ in DNCB-induced mice skin (Figure 6).”
Materials and Methods section
“4.8. Isolation of RNA, cDNA synthesis, and real-time reverse transcription-polymerase chain reaction (RT-PCR) analysis
Total RNA was extracted using Trizol reagent (Invitrogen Life Technologies, CA, USA) following the manufacturer’s instructions. The total RNA was quantified by measurement using a nanodrop2000 spectrophotometer (Thermo scientific, NC, USA). Total RNA was transformed into cDNA using ReversTra AceTM qPCR RT master mix (TOYOBO, Osaka, Japan) according to the following protocol: Incubate at 37 °C for 20 min, incubate at 50 °C for 5 min, heat to 98°C for 5 min, and store the reacted solution at 4°C. Real-time RT-PCR was performed on a CFX384 Real-time system (Bio-rad, CA, USA) by using THUNDER-BIRDTM SYBRTM qPCR Mix (TOYOBO, Osaka, Japan). Primers for PCR amplification were as follows: GAPDH: 5’-AAC GAC CCC TTC ATT GAC-3’/5′- TCC ACG ACA TAC TCA GCA C-3′; IFN-γ: 5′- AGA GGA TGG TTT GCA TCT GGG TCA-3′/5′- ACA ACG CTA TGC AGC TTG TTC GTG-3′; TNF-α: 5′- ATG AGC ACA GAA AGC ATG AT-3′/5′- TAC AGG CTT GTC ACT CGA AT-3′. The RT-PCR reactions were cycled 40 times, denaturation (95 °C, 15 s), annealing (60 °C, 1 min). The fold change in target gene expression relative to control was normalized to -actin using the 2-ΔΔCt method. Real-time RT-PCR was con-tributed to detect the effects of relevant inflammatory factors on mRNA.”
Minor Comments
- Artemisia anomala should be described in italics?
: We corrected it.
- In Fig. 3, the description about EAA dose (second rows in each) should be changed from “-, +, 1, 10, 50” to “-, -, 1, 10, 50”.
: We corrected the Figure 3.
- Since enough margin is provided, the authors should enlarge Fig. 6 for more easy to see.
: We corrected it.
- If the authors have the approved number of the animal experiment, please provide it in the microscopy (40 x or 100 x)
: We inserted the approved number of the animal experiment in material and methods section. Also, we inserted the microscope magnification (40 x) to the results section.
“All procedures for the animal study were approved by the Korea Institute of Oriental Medicine Institutional Animal Care and Use Committee (KIOM-IACUC; D-17-013) and were conducted in accordance with US guidelines (NIH publication #83–23, revised in 1985).”
“(b) The AD dorsal skin lesions were sectioned and stained with H&E (40 x).”

Reviewer 2 Report
Introduction section
The authors must show other studies showing the isolated compounds obtained in the Artemisia anomala ,as well the properties of them as a anti-inflammatory profile
Methods section
The authors must show the intra and inter range assays and the specificity and sensibility of each ELISA kit used in the experiments
Histological analysed must be showed by using scores to the studied elements
Results section
The authors must show the percentage of inhibition of each studied pro-inflammatory mediators.
Discussion section
The results obtained must be related to the other studies have already published.
Figure 4
Legend the authors must show what was the increase of the studied parameters observed in the microscopy (40 x or 100 x)
Author Response
Introduction section
The authors must show other studies showing the isolated compounds obtained in the Artemisia anomala ,as well the properties of them as a anti-inflammatory profile
: We inserted the sentence in Introduction section.
“Also, constituents of Artemisia anomala S. Moore, such as quercetin, apigenin, and tricin, were reported the effects of anti-inflammation, anti-oxidant and anti-cancer.”
Methods section
The authors must show the intra and inter range assays and the specificity and sensibility of each ELISA kit used in the experiments
: As indicated in the legend of Figure 2, triplicates were performed on the same plate for ELISA experiments, and at least three independent experiments were performed.
Histological analysed must be showed by using scores to the studied elements
: The histopathological results of the epidermal thickness of the dorsal skin and the ear skin thickness of the atopic dermatitis induction model using DNCB (Figures 4 and 5a), there was a significant difference visually. It is considered sufficient to prove the effect of EAA in the DNCB-induced AD model.
Results section
The authors must show the percentage of inhibition of each studied pro-inflammatory mediators.
: In general, it is more efficient and intuitive to indicate the quantitative value of each pro-inflammatory mediator rather than indicating the percentage of inhibition of the pro-inflammatory mediators in skin disease studies. Also, most previous studies indicate quantitative values rather than the percentage of inhibition [1 – 3].
- Pyropia Yezoensis Extract Suppresses IFN-Gamma and TNF-Alpha-Induced Pro-inflammatory Chemokine Production in HaCaT Cells via the Down-Regulation of NF-κB. Nutrients 2020, 12, 1238
- Korean Red Ginseng attenuates ultraviolet-mediated inflammasome activation in keratinocytes. J Ginseng Res. 2021, 45, (3), 456–463.
- Bee Venom Inhibits Porphyromonas gingivalis Lipopolysaccharides-Induced Pro-Inflammatory Cytokines through Suppression of NF-κB and AP-1 Signaling Pathways. Molecules 2016, 21, (11), 1508.
Discussion section
The results obtained must be related to the other studies have already published.
: We inserted the sentence in Discussion section.
“In this experiment, we elucidated that EAA inhibits the production of pro-inflammatory cytokines and chemokines, such as IL-6, IL-8, RANTES and TARC in TNF-α/IFN-γ-stimulated keratinocytes in consistent with previously reports [20,21].”
Figure 4
Legend the authors must show what was the increase of the studied parameters observed in the microscopy (40 x or 100 x)
: We inserted the microscope magnification (40 x) to the results section.
“(b) The AD dorsal skin lesions were sectioned and stained with H&E (40 x).”

Round 2
Reviewer 1 Report
I think that the submitted manuscript has been satisfactory revised by the authors.
Author Response
Thank you for your kindness.
Reviewer 2 Report
The authors did not modified the manuscript as recommended. Its important to obtain scores to histologic analysis to statistical analysis. Also percentage of inhibition show better the results. The intra and inter assay must be showed in the cytokines kits analysis.
Author Response
- The authors must show the percentage of inhibition of each studied pro-inflammatory mediators.
: We inserted the sentence in Results section.
“As shown in Figure 2, EAA significantly reduced the production of cytokines and chemokines such as RANTES, IL-8, TARC, and IL-6 in HaCaT cells co-stimulated by TNF-α and IFN-γ in a concentration-dependent manner. EAA treatment significantly decreased the productions of RANTES, IL-8, TARC, and IL-6 to 21%, 22.5%, 24.5%, and 8.2% at 50 µg/ml, respectively.”
- Histological analysed must be showed by using scores to the studied elements
: We inserted the sentence in Materials and methods section, and added the result of epidermal thicknesses in Results section.
Materials and methods section
4.7. Histopathological observation
In order to reduce the error, the designated researcher continuously measured. Ear thickness and spleen size were evaluated by micrometer (Mitutoyo, Kawasaki, Japan). For histological analysis, Dorsal skins were fixed in 4% paraformaldehyde and embedded in paraffin wax. The skin tissue sections were stained with hematoxylin and eosin (H&E) staining to measure changes in epidermal thicknesses. All skin images were acquired under Nikon Eclipse Ti microscope (Nikon, Tokyo, Japan). Representative sections of the epidermis and dermis by H&E staining (magnification ×400, scale bar: 100 μm). The thickness of skin epidermis was measured from each five locations using Image J software (National Institute of Health, Starkville, MD, USA).
Results section
Figure 4. Inhibitory effects of EAA on DNCB-induced AD in vivo model. Representative photo-graphs of dorsal skin lesions and histopathological observations in DNCN-induced AD mice with vehicle (solvent), control (DNCB-induced AD), EAA (50 and 100 mg/kg) and dexamethasone (10 mg/kg) via oral administration. (a) Dorsal skin photographs taken at the end of the experi-ment showing AD‐like skin lesions. (b) The AD dorsal skin lesions were sectioned and stained with H&E (magnification ×40, scale bar: 100 μm). (c) Measurement of the epidermal thickness. Bars with different letters indicate statistically significant difference at *p < 0.05 and ***p < 0.001 compared with the control group (solvent). - The intra and inter assay must be showed in the cytokines kits analysis.
: First of all, we apologize for not being able to meet all of your requirements due to time constraints. We determined the ELISA intra-assay precision in three individual samples by calculating the standard deviations and the coefficient of variation. In the next experiment, we will proceed with the experiment based on your comments.
